# Pathogenic Mitochondria DNA Mutations: Current Detection Tools and Interventions

**DOI:** 10.3390/genes11020192

**Published:** 2020-02-12

**Authors:** Mohd Fazirul Mustafa, Sharida Fakurazi, Maizaton Atmadini Abdullah, Sandra Maniam

**Affiliations:** 1Department of Human Anatomy, Faculty of Medicine and Health Sciences, Universiti Putra Malaysia, Selangor Darul Ehsan 43400, Malaysia; mfazirul@gmail.com (M.F.M.); sharida@upm.edu.my (S.F.); 2Department of Pathology, Faculty of Medicine and Health Sciences, Universiti Putra Malaysia, Selangor Darul Ehsan 43400, Malaysia; maizaton@upm.edu.my; 3Laboratory of Molecular Medicine, Institute of Bioscience, University Putra Malaysia, Selangor Darul Ehsan 43400, Malaysia

**Keywords:** mitochondrial DNA, mitochondria DNA mutations, mitochondrial DNA diseases, mitochondria transfer, genetic intervention

## Abstract

Mitochondria are best known for their role in energy production, and they are the only mammalian organelles that contain their own genomes. The mitochondrial genome mutation rate is reported to be 10–17 times higher compared to nuclear genomes as a result of oxidative damage caused by reactive oxygen species during oxidative phosphorylation. Pathogenic mitochondrial DNA mutations result in mitochondrial DNA disorders, which are among the most common inherited human diseases. Interventions of mitochondrial DNA disorders involve either the transfer of viable isolated mitochondria to recipient cells or genetically modifying the mitochondrial genome to improve therapeutic outcome. This review outlines the common mitochondrial DNA disorders and the key advances in the past decade necessary to improve the current knowledge on mitochondrial disease intervention. Although it is now 31 years since the first description of patients with pathogenic mitochondrial DNA was reported, the treatment for mitochondrial disease is often inadequate and mostly palliative. Advancements in diagnostic technology improved the molecular diagnosis of previously unresolved cases, and they provide new insight into the pathogenesis and genetic changes in mitochondrial DNA diseases.

## 1. Mitochondria as the Energy Source in Cells

Historically mitochondria evolved from a bacterial ancestor of α-proteobacteria and became endosymbionts living inside eukaryotes over one billion years ago [1]. Under normal physiological conditions, mitochondria produce most of the adenosine triphosphate (ATP) through the oxidative phosphorylation system (OXPHOS). The OXPHOS is composed of five protein complexes (complexes I–V), and mitochondria DNA (mtDNA) only encodes 13 structural subunits of complex I, III, IV, and V, whilst complex II is completely nuclear encoded. During OXPHOS, mitochondria produce reactive oxygen species (ROS) known as mitochondrial ROS (mtROS), which are formed as a consequence of proton leak during respiration at the inner mitochondrial membrane. The mtROS formed increases the risk of mtDNA perturbation and impairment of ATP synthesis, and it contributes to overall mitochondrial dysfunction [2].

Mitochondria are dynamic, constantly fusing and dividing [3]. A major component of the cellular control of mitochondria integrity is the specialized form of autophagy known as mitophagy. Mitophagy is a highly specific form of autophagy which degrades dysfunctional or excessive mitochondria through the process of the autophagasome–lysosomal system [4]. It targets mitochondria for degradation at the autophagosome through interaction of key adaptor molecules at the outer mitochondrial membrane. These adaptor molecules include B-cell lymphoma 2 (Bcl-2/adenovirus E1B 19 kDa protein-interacting protein 3 (BNIP3), Nip3-like protein X (NIX), and function unknown now (FUN)14 domain-containing 1 (FUNDC1), in addition to mitochondrial targets of E3 ubiquitin ligases functioning at the mitochondria, such as Parkin and Mul1 [4].

## 2. Genetics Insight of Mitochondria

Mitochondria are unique compared to other organelles in animals as they have their own DNA. The mtDNAs are arranged in protein–DNA complexes that are also known as mitochondrial nucleoids which are found in the mitochondrial matrix [5]. Human mtDNA contains 16,569 base pairs of nucleotides and encodes for 37 genes which are maternally inherited [6]. The two strands of mtDNA are differentiated by their nucleotide content; the purine-rich strand is known as the heavy strand (H-strand), while the pyrimidine-rich strand is known as the light strand (L-strand) [7]. The H-strand encodes 28 genes, and the L-strand encodes nine genes [7]. Of the 37 genes, 13 genes encode OXPHOS subunits, 22 encode transfer RNAs (tRNAs), two encode ribosomal RNAs (rRNAs), and less than 10% of the genome is non-coding [7]. Thus, the mitochondrial genome has a highly compact genome.

MtDNA is packaged into large nucleoprotein complexes named nucleoids, and the major structural component of the nucleoid is transcription factor A, mitochondrial (TFAM). TFAM binds to DNA without sequence specificity and is essential in mitochondrial transcription machinery [8]. The non-coding region (NCR) in the mtDNA harbors the origin of H-strand DNA replication, while the origin for L-strand replication is located outside the NCR. MtDNA transcription is directed by a cis-acting element. It is located at the 5′ boundary of the regulatory sequence known as the displacement loop (D-loop) region, which also acts as the starting point for replication to occur [9]. Several polymerases such as PrimPol [10], DNA polymerase β [11], DNA polymerase θ, DNA polymerase ζ [12], and DNA polymerase γ [13] are reported to play a role in mitochondria. However, DNA polymerase γ is the replicative polymerase in mitochondria [14]. Once DNA polymerase γ completes synthesis, the newly formed DNA strands are ligated by DNA ligase III. The replication of mtDNA is observed during embryogenesis, from fertilized oocyte throughout the process of pre-implantation embryos [15]. Most of the proteins present in mitochondria are encoded by nuclear DNA. MtDNA encodes the full set of tRNAs and the two ribosomal RNAs required for mitochondrial translation. To date, the mechanism of mitochondrial translation is poorly understood. Dysfunction in mitochondrial replication results in single or multiple mtDNA mutations which lead to deletion or depletion of mtDNA.

## 3. MtDNA Diseases

MtDNA diseases result in severe multisystem complications during the neonatal phase, adolescence, or, in some cases, during adult onset [16] with a prevalence of one in 4300 [17]. Initially, “mtDNA diseases” were used as a collective term for a heterogeneous group of genetic disorders characterized by defective oxidative phosphorylation. However, with the remarkable progress in understanding the role of mitochondria—and of nuclear-derived proteins in mitochondrial function—the definition also includes defects in the lipid milieu, in mitochondrial translation, and in mitochondrial fission and fusion [18,19,20].

MtDNA disorders were initially considered to be uncommon due to the widespread symptoms which involve cardiovascular, neurological, and age-related degenerative diseases. Patients with mitochondrial disease display a cluster of clinical features that can be categorized into a discrete clinical syndrome. Common mitochondrial diseases include Leigh syndrome, Leber hereditary optic neuropathy (LHON), myoclonic epilepsy with ragged red fibers (MERRF), and mitochondrial encephalomyopathy, lactic acidosis, and stroke-like episodes (MELAS) (Figure 1, Appendix A).

The first population-based study of a single pathogenic mtDNA mutation was reported in Finland with the prevalence of MELAS as high as 16.3 in 100,000 [21]. In England, two studies on mitochondrial disease were conducted in the early 2000s. The first study reported on the prevalence of patients with mitochondrial disease or the risk of developing disease in a northern England population, which was 12.78 in 100,000 [22], and the second reported on the prevalence of LHON mutations within a northeast England population, which was 11.82 in 100,000 [23]. The prevalence of LHON was also reported in the Dutch and Finnish populations as 2.6 in 100,000 and 2.06 in 100,000, respectively [24,25]. A prevalence study in an Asian population reported on the prevalence of MELAS as 0.2 in 100,000 [26].

### 3.1. Mitochondrial Encephalomyopathy, Lactic Acidosis, and Stroke-Like Episodes (MELAS)

MELAS is one of the most frequently maternally inherited mitochondrial disorders, and it impairs mitochondrial translation and protein synthesis, which results in the inability of mitochondria to meet the energy demand of various organs, eventually causing multi-organ dysfunction [27]. MELAS is diagnosed and characterized by strokes with hemiparesis and hemianopsia. It usually affects individuals aged under 40 years and it was also observed in patients during childhood. Studies showed that more than 80% of patients with MELAS feature the m.3243A>G mutation in the mitochondrially encoded tRNA leucine 1 (*MT-TL1*) gene [28,29]. Other mutations identified in MELAS include m.3271T>C [30] and m.1642G>A in the mitochondrially encoded tRNA valine (*MT-TV*) gene [31], which is a protein-encoding gene, m.9957T >C in the mitochondrially encoded cytochrome C oxidase III (*MT-CO3*) gene [32], and several mitochondrially encoded reduced nicotinamide adenine dinucleotide (NADH) ubiquinone oxidoreductase chain 5 (*MT-ND5*) mutations (m.1277A>G, m.13045A>C, m.13513G>A, and m.13514A>G) [33,34,35,36]. These mutations subsequently lead to destabilization of tRNA which results in reduction of OXPHOS proteins and insufficiency of complexes I, III, and IV [28,37].

### 3.2. Myoclonic Epilepsy with Ragged Red Fibers (MERRF)

MERRF syndrome is a rare syndromic mitochondrial disorder (MID) with broad phenotypic but narrow genotypic heterogeneity. Commonly, it is known as a chronic neurodegenerative disease that can manifest in both children and adults. Symptoms of MERRF include myoclonus, myopathy, and spasticity. Mutations in the mitochondrially encoded tRNA lysine (*MT-TK*) gene (m.8344A>G, m.8356T>C, m.8363G>A) are the most common cause of MERRF, occurring in more than 80% of all cases [38]. Mutations were also reported in the *MT-TL*/tRNA (Leu) (m.3291T>C) [39,40], tRNA (Ile) (m.4279A>G) [41], *MT-TF*/tRNA (Phe) [42], and *MT-TP*/tRNA (Pro) genes [43].

### 3.3. Leber Heridetary Optic Neuropathy (LHON)

LHON patients typically present with painless loss of central vision in one eye, followed by loss of vision in the second eye within weeks or months [44]. It predominantly affects males (80%) with disease onset between 15 and 30 years [45]. The primary cause of this disease is a mutation of the mtDNA, with a single amino-acid substitution in one of the mtDNA-encoded subunits of NADH ubiquinone oxidoreductase, the first complex of the electron transport chain. The majority of LHON cases are caused by single-nucleotide point mutations of mtDNA located in NADH dehydrogenase subunit 1 (*ND1*) (G3460A), *ND4* (G11778A), or *ND6* (T14484C) genes, which result in a dysregulated complex I of OXPHOS [46].

### 3.4. Leigh Syndrome

Leigh syndrome, a highly heterogeneous disorder and the most common pediatric mitochondrial disease characterized by progressive neurodegenerative disorder, is caused by mutations in almost 80 different genes [18]. It is a rare inherited subacute necrotizing encephalomyelopathy that affects the central nervous system, and the onset of symptoms is typically seen between the ages of three and 12 months, often following a viral infection. MtDNA-associated Leigh syndrome is often seen in the neonatal phase. Several mutations of nuclear genes that affect assembly factors or subunits of the mitochondrial respiratory chain, mtDNA replication, transcription, and translation, as well as proteins involved in other mitochondrial processes, like pyruvate metabolism, coenzyme Q10 biosynthesis, and the oxidation of fatty acids, and non-mitochondrial processes, like thiamine metabolism, which affects mitochondrial function, result in “classic” Leigh syndrome or Leigh-like syndrome [47].

### 3.5. Other mtDNA Diseases

MtDNA deletion syndromes are caused by a single large-scale deletion in the mtDNA genome, which include diseases such as Pearson syndrome, Kearns–Sayre syndrome (KSS), and chronic progressive external ophthalmoplegia (CPEO) [48]. Pearson syndrome is a very rare syndrome characterized by bone marrow failure, severe transfusion-dependent sideroblastic anemia, and variable exocrine pancreatic insufficiency [49]. Death may occur in early infancy, or survival after recovery from bone marrow dysfunction is possible, with a transition to clinical manifestations of KSS. KSS is a rare neuromuscular disorder known as mitochondrial encephalomyopathy with a prevalence of 1–3 cases in 100,000 [50], presented before the age of 20 years. One of its features is PEO, a disorder that affects children before the age of 10 with limited eye movements, bilateral ptosis, and orbicularis weakness. PEO is generally associated with single mtDNA deletions and considered the least lethal among the three syndromes [51]. MtDNA dysfunction in diabetes is known as maternally inherited diabetes and deafness (MIDD), which was first reported in 1992. A single point mutation in the mtDNA affects the activities of complex I and IV in the respiratory chain, which results in cellular energy deficiency in metabolically active organs such as the pancreas and cochlea [52]. Other mtDNA diseases with low prevalence include neurogenic muscle weakness, ataxia, and retinitis pigmentosa (NARP).

## 4. Detection of Mutations in mtDNA

Methods for detecting mutations in mtDNA do not differ much from those used to determine the primary sequence of any DNA. The first generation of DNA sequencing involved fragmentation and detection of radiolabeled DNA, and these protocols quickly improved over the years with parallel sequencing techniques [53]. In the past decade, the genetic approach widely used to confirm mtDNA disorders is next-generation sequencing (NGS) [54]. NGS utilizes single-stranded DNA fragments, which results in high background error frequency. A more sensitive approach, duplex sequencing (DS), allows >10,000-fold greater accuracy than conventional next-generation sequencing (NGS) [55]. DS sequences both strands, and it scores mutations that are present as complementary substitutions in both strands of a single DNA molecule as compared to NGS. The first report on DS was a study investigating the mutational variations of the whole mitochondrial genome between non-stem cells and stem cells in human breast tissues. The study reported that the mitochondrial genome of stem cells has a lower mutation burden compared to non-stem cells, and that the majority of mutational variations occurred randomly [56].

Recently, rolling circle amplification (RCA) was used to amplify mtDNA in a heritability analysis study of 35 individuals [57]. RCA, a low sequence-dependent amplification, was used to amplify the circular mitochondrial genome in a single reaction without the use of primers and temperature regulation, and this sequencing strategy was described as mitochondrial DNA analysis by rolling circle amplification and sequencing (MitoRS). This method was concluded to be a robust, accurate, and sensitive analysis, suitable for large samples [57].

## 5. MtDNA Intervention

This review outlines the current advances in in vitro mitochondria manipulation, which allows researchers to understand the pathogenic implication and therapeutic potential of mtDNA mutation. The strategies involve either removing the detrimental mtDNA by transferring healthy mitochondria or targeting specific mtDNA sequences that cause mitochondrial disease (Figure 2).

Mitochondria transfer technologies focus on introducing exogenous mitochondria to a recipient cell. In this technique, the mtDNA is not manipulated. MtDNA replacement technology is a more specific and targeted method that can generate non-native mtDNA sequences or repair the sequences of existing mtDNA to shift the mtDNA heteroplasmy ratio.

### 5.1. Artificial Mitochondria Transfer

The mitochondrion is an endosymbiotic organism with partial nuclear independence, allowing it to be exchanged between cells [58]. The transfer of exogenous mitochondria to an existing endogenous mitochondrial network can lead to an alteration in the function and bioenergetic profile of the recipient cells. Successful intercellular mitochondrial transfer relies on the communication between cells, which includes membrane nanotubes and other cytoplasmic bridges, exosomes, and mitochondrial fusion–fission mechanisms [59].

Exogenous mtDNA molecules are either incorporated into mammalian cells or into empty mitochondria and then are introduced into mammalian cells via endocytosis (Patent No: EP3067416A1). Viable isolated mitochondria can be internalized by simple co-incubation, which involves macropinocytosis. However, the internalization is only for a short duration of time [60]. Another quick and simple method is via centrifugation, which is also suitable for all cell types to transfer viable mitochondria into target cells. A non-ionic surfactant, PF-68, is used to enhance cell permeability, which increases the number of mitochondria penetrating the cell membrane [61]. MitoCeption is a method for quantitatively transferring mitochondria from human mesenchymal stromal/stem cells (MSCs) to cancer cells. Various types of stromal and cancer cell communications were documented, which include cytokine-dependent, metabolite exchange, and direct cell–cell contacts [62]. MitoCeption transfers mitochondria isolated from MSCs to cancer cells by simple coculture, and, at the end, cancer cells contain both endogenous and exogenous mitochondria [63]. It was reported that this transfer resulted in an increase in mtDNA concentration, OXPHOS activity, and ATP production of the cancer cells [58].

### 5.2. Genetic Transfer to the Mitochondria

The mitochondrial inner membrane that selectively transports molecules into the mitochondrial matrix is impermeable to hydrophilic molecules. Exogenous genes with functional protein expression are transported into mitochondria to compensate for mitochondrial dysfunction as a consequence of mtDNA mutations.

Labeled DNA oligonucleotides are transferred into the mitochondrial matrix using peptide nucleic acid (PNA) [64]. PNA has a similar structure to DNA or RNA; however, the sugar phosphate backbone is replaced by a peptide backbone. Labeled DNA oligonucleotides are introduced into the mitochondrial matrix using PNAs conjugated to mitochondrial-targeting peptides as a vehicle to be transported through the translocase outer membrane (TOM)/ translocase inner membrane (TIM) import apparatus.

Biolistic transformation utilizes a helium shockwave to deliver DNA on microscopic metal particles that is incorporated into mtDNA via active homologous recombination [65]. To date, single-cell eukaryotes such as *Saccharomyces cerevisiae* [65] and *Chlamydomonas reinhardtii* [66] are the only species where biolistic transformation was used to deliver DNA into an organelle.

### 5.3. MtDNA Gene Editing

Gene editing in the mitochondria is based on the concept of using the inefficient double-strand break repair system in mitochondria and introducing endonucleases to degrade the mutant mtDNA and to repopulate with wild-type mtDNA. Pathogenic mtDNA mutations are generally heteroplasmic with the presence of pathology characteristics when the ratio of mutated mtDNA exceeds a certain threshold.

Recently, Gammage et al. improved their previous findings on zinc finger nucleases (ZFN) to target and cleave predetermined loci in mtDNA [67]. They generated a mitochondrially targeted ZFN that carries two cleavage domains linked to the same protein that selectively eliminates pathogenic mtDNA, as well as a region of mtDNA that is most frequently associated with diseases, which contains several transfer RNAs and structural genes of the OXPHOS apparatus [68].

Mitochondrially targeted transcription activator-like effector nucleases (mitoTALENs) are used to cleave specific sequences in mtDNA with the goal of eliminating mitochondria carrying pathogenic point mutations. This approach was recently explored in two clinically important mtDNA point mutations associated with mitochondrial disease, which include myoclonus epilepsy with ragged red fibers (MERRF) and MELAS/Leigh syndrome [69]. The mutation load was successfully reduced in vitro, as depicted by improved biochemical oxidative phosphorylation defects. The authors described the location of the mutations and the size of the mitoTALEN as being the major challenge in translating the mitoTALEN approach into a clinical setting [69].

MitoTALEN specific for the mtDNA region harboring the m.5024C>T mutation was cloned into an adeno-associated virus and phage (AAVP) vector to test its role in regulating mtDNA heteroplasmy in vivo [70]. A mouse with heteroplasmic mitochondrial tRNA^Ala^ gene mutations was the first mouse model with a heteroplasmic pathogenic mtDNA mutation generated. This mouse is associated with tRNA^Ala^ instability and a mild cardiac phenotype at old age [70]. It resembles human heteroplasmy mtDNA mutations in tRNA^Ala^ characterized by cytochrome C oxidase (COX)-deficient fibers that are associated with myopathy and impairment of OXPHOS [71]. A significant decrease in mutant/wild-type ratio expressed in skeletal and cardiac muscle compared to non-targeted tissue was observed after systemic delivery of AAV9–mitoTALEN [70].

## 6. Mitochondrial Manipulation in Clinics

Research and trial of mitochondrial manipulation raise several bioethical issues. The clinical use of mitochondrial modification involves germline modification. Unlike somatic modification, these manipulated genes are transmitted to the offspring. The ability to freely manipulate genes hinders the future generation from receiving an unmanipulated gene pool. Furthermore, children born from a mitochondrial replacement procedure would have a genetic link to three people: their parents and the donor [72,73].

Predictive tests such as pre-implantation genetic diagnosis (PGD) and pre-natal diagnosis (PND) are available to avoid mtDNA disorder transmission. The two reproductive options that are well documented for mitochondrial donation in women with pathogenic mtDNA mutations who wish to reduce the risk of mitochondrial disorder in their children are maternal spindle transfer (MST) in unfertilized oocytes and pronuclear transfer (PNT) in zygotes [74]. The first country to allow the clinical use of mitochondrial transfer under license was the United Kingdom in October 2015 [75]. This was followed by the United States (US) in February 2016 with restricted use [76]. The panel of US experts recommended mitochondrial transfer to only be used to generate male babies to prevent the transmission of surrogate donor mitochondria to the future generation [77]. The first successful clinical case of reduced maternal mutated mtDNA transfer using MST resulting in the birth of a healthy boy was reported in 2017 [78]. The female carrier was asymptomatic and carried an 8993T>G mtDNA mutation in the *MT-ND6* gene, associated with Leigh syndrome [78].

The debate about the ethics of germline modification is inevitable. The manipulation techniques which involve egg donation remain controversial and pose a potential risk to the donor. Since the transmission of mtDNA disorders is complex and hard to predict, a safe and effective technique with appropriate information and support is pertinent to the affected patient. Significant precaution measures should be taken in techniques involving altering genes; however, the goal of treatment is to alleviate suffering of the patient, and their families should also be considered.

## 7. Conclusions and Future Perspectives

MtDNA disorders are among the most common inherited human diseases, and their prevalence was shown to be influenced by demographic and genetic factors [79]. The heterogeneity of the mitochondrial genome poses unmet challenges to researchers. The in vitro modeling of mtDNA diseases is not straightforward, as it suffers additional complications due to the heteroplasmy and homoplasmy state of mitochondrial mutations. Moreover, the significant contribution of nuclear-encoded genes in regulating the mutation effect complicates the model even further. Despite these limitations, in vitro models are important in establishing the cause of mtDNA diseases. Initially, specific mtDNA sequences involved in the disease are identified and analyzed. The detrimental mtDNA sequences are either altered via gene editing or transferred into a different nuclear background, which allows the investigators to explore the potential effect of the specific mtDNA sequences on mitochondrial function and cell metabolism. The recent genomics advancements established the molecular diagnosis of suspected mtDNA diseases.

Several mtDNA diseases are due to either dysfunction in mitochondrial replication or translation, which results in mtDNA mutation(s) or a mutation in nuclear-encoded genes that regulate mitochondrial pathways involved in ATP generation, encoding subunits of OXPHOS and associated assembly factors, enzymes involved in ubiquinone biosynthesis, and the pyruvate dehydrogenase complex. Despite the major advances highlighted in this review, current available treatment options for mtDNA diseases are limited and focus on disease management. However, assisted reproductive techniques which allow almost complete replacement of the cytoplasm of egg/embryo, eliminating the undesired mutated mitochondria, are proposed to families with mtDNA mutations that are transmitted down the maternal lineage. The development of techniques that genetically manipulate the detrimental mtDNA sequences either in vitro or in vivo may provide potential treatment for mtDNA diseases.

## Figures and Tables

**Figure 1 genes-11-00192-f001:**
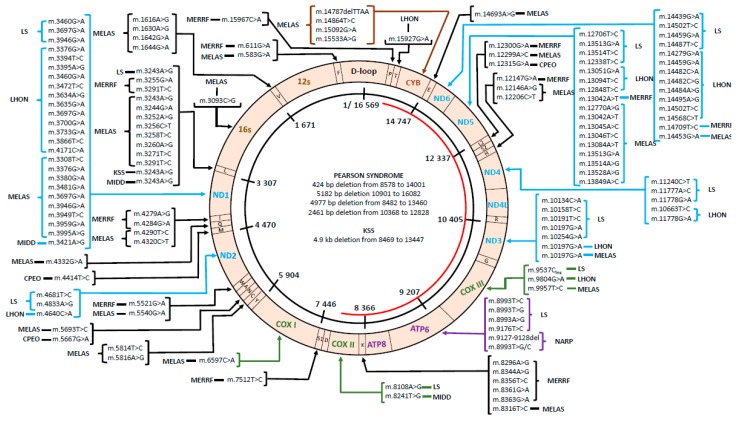
Mitochondrial DNA (mtDNA) mutations reported in different types of diseases. Mitochondrial genome diagram and common mutations reported in human diseases. Mutations are depicted by locations of mutated base, and the single large-scale deletions are shown in the center of the genome. (C: cytosine, G: guanine, T: thymine, A: adenine, ins: insertion, del: deletion). The D-loop, a non-coding region; ND1, reduced nicotinamide adenine dinucleotide (NADH) ubiquinone oxidoreductase chain 1; ND2, NADH ubiquinone oxidoreductase chain 2; COX I, cytochrome oxidase I; COX II, cytochrome oxidase II; ATP 8, ATP synthase 8; ATP 6, ATP synthase 6; COX III, cytochrome oxidase III; ND3, NADH dehydrogenase 3; ND4L, NADH ubiquinone oxidoreductase chain 4L; ND4, NADH dehydrogenase 4; ND5, NADH dehydrogenase 5; ND6, NADH dehydrogenase 6; CYB, cytochrome b.

**Figure 2 genes-11-00192-f002:**
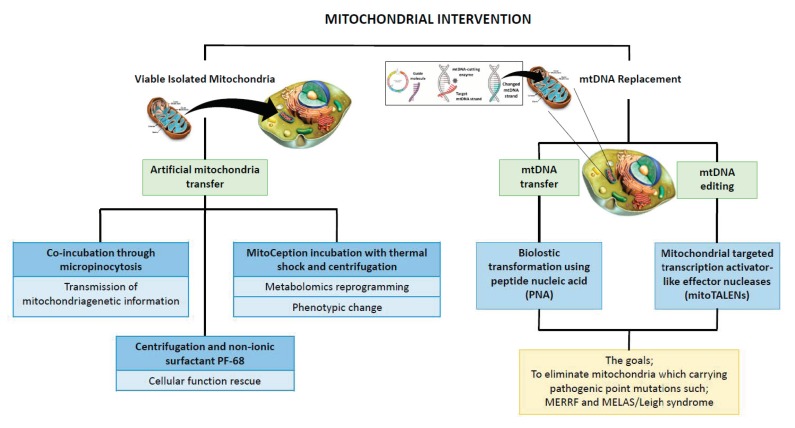
Advances in the past decade in manipulating mitochondria genetic content.

## Data Availability

The datasets analyzed during the current study are available from the corresponding author on reasonable request.

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
