# Peer review of "Pathogenic Mitochondria DNA Mutations: Current Detection Tools and Interventions"

_genes, 2020, doi:10.3390/genes11020192_

Round 1
Reviewer 1 Report
In this review, the authors first summarized the main diseases caused by point mutations in mtDNA and how these mutations can be detected. They then described the techniques used in vitro to manipulate mtDNA and briefly cite the use of mitochondrial donation for women carrying pathogenic mtDNA mutations.
As a general comment this review lacks figures (there is only one table in the suplementary data) to illustrate the text.
Concerning the text, the description of the different diseases caused by mtDNA mutations is really not new and could be avoided or reduced to a minimum (paragraph 3). On the opposite, even if the title mention"current detection tools", only a short paragraph is dedicated to this interesting aspect (line 140 to 158). This paragraph is too short, and misses a lot of information, for instance it doesn't mention RNAseq or the different options for NGS (whole genome sequencing, targeted genes etc.)
Overall I find that this review is too generic, with a lot of information missing and lacks illustrations.
Author Response
Reviewer 1
In this review, the authors first summarized the main diseases caused by point mutations in mtDNA and how these mutations can be detected. They then described the techniques used in vitro to manipulate mtDNA and briefly cite the use of mitochondrial donation for women carrying pathogenic mtDNA mutations.
Comment 1
As a general comment this review lacks figures (there is only one table in the suplementary data) to illustrate the text.
We have included two new figures that depict the various mitochondrial mutations (Figure 1) as well as a figure to summarize the tools used in mtDNA intervention (Figure 2).
Comment 2
Concerning the text, the description of the different diseases caused by mtDNA mutations is really not new and could be avoided or reduced to a minimum (paragraph 3).
It is almost impossible not to mention the common mtDNA mutation diseases such as MELAS, LHON, MERRF and Leigh Syndrome when mtDNA mutations are being discussed. We have improved this section by including other mtDNA diseases which are rare but present in the population i.e Pearson syndrome, KSS, CPEO, MIDD and NARP. We are aware of newer mitochondrial diseases such as MEMSA and mitochondrial neuro-gastro-intestinal encephalomyopathy (MNGIE). However, these diseases were not included as they are caused by nuclear gene mutation that results in mitochondrial dysfunction. In this section of the manuscript, we are attempting to highlight the effect of mtDNA mutations and the diseases.
Comment 3
On the opposite, even if the title mention"current detection tools", only a short paragraph is dedicated to this interesting aspect (line 140 to 158). This paragraph is too short, and misses a lot of information, for instance it doesn't mention RNAseq or the different options for NGS (whole genome sequencing, targeted genes etc.)
We acknowledge and agree that there are many options available to investigate the mitochondrial genome. Technique such as RNA-seq was not mentioned as this is used to measure the RNA from the genome and not direct sequencing of the mtDNA. Similarly, to our knowledge we did not find any reports on targeted sequencing in mitochondria. This manuscript attempts to explain the detection tools in the last 10 years that are used to identify specific mutations in the mtDNA that results in mitochondrial diseases.
Overall I find that this review is too generic, with a lot of information missing and lacks illustrations.

Reviewer 2 Report
Review of “Pathogenic Mitochondrial DNA mutations: Current Detection Tools and Interventions”
This manuscript is a review of the mutations on human mtDNA that cause disease and the current state-of-the-art techniques that are used to diagnose and treat these diseases. This ground has been covered by other reviews, but the addition of some of the most recent molecular intervention techniques makes the review useful and interesting. Table S1 is particularly useful with a compilation of mitochondria mutations and their location. Having said this, the manuscript can be significantly improved in two ways: [1] the English needs to be improved. In most cases it can be deduced what the authors are trying to say but in some cases the exact meaning is lost. This makes the paper rather hard to read. I have included a number of corrections that I strongly recommend be made before publication. [2] The references in the paper and in Table S1 do not have volume numbers and page numbers! The references are the heart of a review and this road block to looking up these references greatly decreases the value of the review. This must be fixed before publication.
Suggested Corrections with explanation.
Line 14: …. Are the only organelles in animals that contain their own genomes. [mitochondria are plural; you must add “ in animals” to make the sentence true because of chloroplasts].
Line 14: … mutation rate of the mitochondrial genome.
Line 16/17: … Pathogenic mitochondrial DNA mutations result in mitochondrial disorders, which are among the most…. [Remove The and make mutations plural, make mitochondrial an adjective, add “which are among” in place of “one of”].
Line 18: Intervention of mitochondrial disorders involve either……..
Line 21: …mitochondrial disease intervention. Although it has been…. [add “disease”, substitute Although for Albeit (archaic)].
Line 23/24: Advancements in diagnostic technology have…..
Line 30/31: Mitochondria are thought to have evolved from an endosymbiotic relationship between an ancestor of α-proteobacteria and a pre-eukaryotic organism to become an endosymbiont inside eukaryotes…… [between bacterial ancestor and …what? In your sentence you use “between” which requires two organisms only one of which is stated].
Line 33: …. The OXPHOS system is composed of five…..
Line 34: …all of them have some subunits encoded by mitochondrial DNA….. [Your sentence makes it sound like all the subunits of the complexes are encoded by mtDNA, which is not true].
Line 35: …, which is completely nuclearly encoded. [delete genome (nDNA)].
Line 36: …(ROS) known as…. [deleted extra “as”].
Line 39: contributes to overall mitochondrial dysfunction. [remove “the”].
Line 40: Mitochondria are dynamic, constantly fusing and dividing. [3]
Line 42: specialized [not specialised (British spelling).
Line 49: … other organelles in animals [in animals added to make sentence true because of chloroplasts].
Line 51: Human mtDNA cvontains….. [[Remove “It has been reported in”]
Line 54: Change “Whilst” (British, archaic) to “While”.
Line 56/57: … of the genome is noncoding.
Line 57: Thus, the mitochondrial genome has a highly compact genome. [replace “consists of” with “has a “]
Line 58: ….into large nucleoprotein complexes called nucleoids
Line 61: Change “Whilst” (British, archaic) to “While”.
Line 62: ….by a cis-acting element.
Line 63: …known as the displacement loop (D-Loop) region which also acts as the….
Line 69/70: [?] Interestingly, no reports on mutations of mitochondrial transcription gene were observed in mitochondrial disease. [I do not understand what is meant by this sentence. What is a “mitochondrial transcription gene”? Do the authors mean a mtRNA Polymerase gene? (Which is not located on mtDNA.) Do they mean tRNA and rRNA genes? (Which would make the sentence false.)]
Line 70/71: Most of the proteins present in mitochondria are [not is] encoded…..
Line 73: Dysfunction in mitochondrial replication results in…. [remove the word “process”].
Line 76/77: … during the neonatal phase or adolescence or, in some cases, during adult onset.
Line 77: … 1 in 4300 [or 4,300].
Line 78: “mitochondrial diseases” was used as a collective term for a heterogeneous group of genetic disorders characterized by…..
Line 80: ….and of nuclear-derived proteins.
Line 81: …in the lipid milieu, in mitochondrial translation, and in mitochondrial fission and fusion.
Line 85: ….features that can be categorized into……
Line 92: … or the risk of developing disease…..
Line 94: ….population as 11.82 per 100,000.
Line 95: …. population as 2.6 per 100,000.
Line 96 … on the prevalence of MELAS as 0.2 per 100,000.
Line 101: …. Various organs….
Line 102: …. It usually affects individuals [not It is usually affects…]
Line 106: Add space before [32].
Line 113: can manifest in both children and adults. [not manifests].
Line 113: Symptoms… include [not includes].
Line 122: a mutation of the mtDNA, a single amino acid substitution in one of the ….. [remove “which causing”, “substitution” is better than “exchange”]
Line 123: The majority of….
Line 128: pediatric [not paediatric (British)]
Line 137: …. which affects mitochondrial function, result in classic [not “results” (mutations result, not mutations results)]
Line 140: Mutations…. do not differ….. [ not mutations does not differ).
Line 144: NGS utilizes single stranded DNA [“utilizes” not “sequences”].
Line 146: DS allows >10,000-fold greater accuracy than conventional NGS [not 10,000-fold more accurate than conventional NGS].
Line 149/150: awkward sentence. I am not sure of the intent. Original sentence: “The first report on DS was a study to compare the mitochondrial genome mutation indentified in breast cancer epithelial cells with stem cell features and without.” Do they mean “The first report on DS was a study to compare the mitochondrial genome mutation indentified in breast cancer epithelial cells with and without stem cell features.”?
Line 151/152: …. and that the majority of mutational variation occurred randomly.
Line 157/158: This method was concluded to be a robust, accurate and sensitive analysis, suitable for large samples.
Line 166: The transfer of exogenous [rather than “artificial”] mitochondria. Space before [53].
Line 167: mitochondrial network can lead to an alteration in the function…
Line 170: space before [54].
Line 172: into empty mitochondria. [remove “shells”].
Line 174: …only for a short duration of time.
Line 177: … permeability which increases the number of mitochondria penetrating….
Line 180: Various types…. Which include …..
Line 183/184: an increase in mitochondrial DNA concentration, OXPHOS activity, and ATP production ….
Line 195: Biolistic transformation utilizes a helium shockwave to deliver DNA on microscopic metal particles….
Line 198/199: …where biolistic transformation has been used to deliver DNA into organelles.
Line 202: … and to repopulate with wild-type DNA.
Line 212: …the goal of eliminating mitochondria …
Line 224: … fibers that are associated with….
Line 253: …. Are among the most common …. [Amongst (archaic)].
Line 256: additional complications due to
Line 257: homoplasmy is misspelled.
Line 264: suspected mitochondrial diseases. [delete “cases”].
Author Response
Reviewer 2
This manuscript is a review of the mutations on human mtDNA that cause disease and the current state-of-the-art techniques that are used to diagnose and treat these diseases. This ground has been covered by other reviews, but the addition of some of the most recent molecular intervention techniques makes the review useful and interesting. Table S1 is particularly useful with a compilation of mitochondria mutations and their location. Having said this, the manuscript can be significantly improved in two ways:
[1] the English needs to be improved. In most cases it can be deduced what the authors are trying to say but in some cases the exact meaning is lost. This makes the paper rather hard to read. I have included a number of corrections that I strongly recommend be made before publication.
The corrections have been made as per suggested.
[2] The references in the paper and in Table S1 do not have volume numbers and page numbers! The references are the heart of a review and this road block to looking up these references greatly decreases the value of the review. This must be fixed before publication.
Reference: We apologise for this technical error. The references are formatted accordingly.
Figure: We have included two new figures that depict the various mitochondrial mutation (Figure 1) as well as a figure to summarize the tools used in mtDNA intervention (Figure 2).
Suggested Corrections with explanation.
1. Line 14: …. Are the only organelles in animals that contain their own genomes. [mitochondria are plural; you must add “ in animals” to make the sentence true because of chloroplasts].
The comment was noted and the word ‘mammalian’ is added.
Mitochondria are best known for their role in energy production and are the only mammalian organelle that contain their own genomes.
2. Line 14: … mutation rate of the mitochondrial genome.
The comment was noted and the sentence was rephrased.
The mitochondrial genome mutation rate is reported to be 10-17 times higher compared to nuclear genomes as a result of oxidative damage caused by reactive oxygen species during oxidative phosphorylation.
3. Line 16/17: … Pathogenic mitochondrial DNA mutations result in mitochondrial disorders, which are among the most…. [Remove The and make mutations plural, make mitochondrial an adjective, add “which are among” in place of “one of”].
The comment was noted and the sentence was rephrased.
Pathogenic mitochondria DNA mutations result in mitochondrial disorders, which are among the most common inherited human diseases.
4. Line 18: Intervention of mitochondrial disorders involve either……..
The comment was noted and the sentence was rephrased.
Intervention of mitochondrial disorders involve either the transfer of viable isolated mitochondria to recipient cells or genetically modifying the mitochondrial genome to improve therapeutic outcome.
5. Line 21: …mitochondrial disease intervention. Although it has been…. [add “disease”, substitute Although for Albeit (archaic)].
The comment was noted and the sentence was rephrased.
…..necessary to improve the current knowledge on mitochondrial disease intervention. Although it has been 31 years since the….
6. Line 23/24: Advancements in diagnostic technology have…..
The comment was noted and the sentence was rephrased.
Advancements in diagnostic technology have improved the molecular diagnosis of previously unresolved cases and provide new insights on the pathogenesis and genetic changes in mitochondrial diseases.
7. Line 30/31: Mitochondria are thought to have evolved from an endosymbiotic relationship between an ancestor of α-proteobacteria and a pre-eukaryotic organism to become an endosymbiont inside eukaryotes…… [between bacterial ancestor and …what? In your sentence you use “between” which requires two organisms only one of which is stated].
The comment was noted and the sentence was rephrased.
Historically mitochondria evolved from bacterial ancestor of a-proteobacteria and became endosymbionts living inside eukaryotes over a billion years ago
8. Line 33: …. The OXPHOS system is composed of five…..
The comment was noted and the sentence was rephrased. The word ‘system’ after OXPHOS was not added to prevent repetition as OXPHOS stands for oxidative phosphorylation system.
The OXPHOS is composed of five protein complexes…..
9. Line 34: …all of them have some subunits encoded by mitochondrial DNA….. [Your sentence makes it sound like all the subunits of the complexes are encoded by mtDNA, which is not true].
The comment was noted and the sentence was rephrased.
The OXPHOS is composed of five protein complexes (complex I–V) and mitochondria DNA (mtDNA) only encodes thirteen structural subunits of complex I, III, IV and V whilst complex II is completely nuclear encoded.
10. Line 35: …, which is completely nuclearly encoded. [delete genome (nDNA)].
The comment was noted and the sentence was rephrased.
The OXPHOS is composed of five protein complexes (complex I–V) and mitochondria DNA (mtDNA) only encodes thirteen structural subunits of complex I, III, IV and V whilst complex II is completely nuclear encoded.
11. Line 36: …(ROS) known as…. [deleted extra “as”].
The comment was noted and the sentence was rephrased.
During OXPHOS, mitochondria produce reactive oxygen species (ROS) known as mitochondrial ROS (mtROS) which are formed as a consequence of proton leak during respiration at the inner mitochondrial membrane.
12. Line 39: contributes to overall mitochondrial dysfunction. [remove “the”].
The comment was noted and the sentence was rephrased.
The mtROS formed increases the risk of mtDNA perturbation, impairment in ATP synthesis and contributes to overall mitochondrial dysfunction.[2]
13. Line 40: Mitochondria are dynamic, constantly fusing and dividing. [3]
The comment was noted and the sentence was rephrased.
Mitochondria are dynamic, constantly fusing and dividing.
14. Line 42: specialized [not specialised (British spelling).
The spelling was corrected.
15. Line 49: … other organelles in animals [in animals added to make sentence true because of chloroplasts].
The comment was noted and the word ‘animals’ was added.
16. Line 51: Human mtDNA cvontains….. [[Remove “It has been reported in”]
The comment was noted and the phrase was removed.
17. Line 54: Change “Whilst” (British, archaic) to “While”.
The comment was noted and addressed.
18. Line 56/57: … of the genome is noncoding.
The comment was noted and addressed.
…..2 rRNAs and less than 10% of the genome is non-coding.
19. Line 57: Thus, the mitochondrial genome has a highly compact genome. [replace “consists of” with “has a “]
The comment was noted and the phrase was replaced.
Thus, the mitochondrial genome has a highly compact genome.
20. Line 58: ….into large nucleoprotein complexes called nucleoids
The comment was noted and addressed.
MtDNA is packaged into large nucleoprotein complexes named nucleoids and the….
21. Line 61: Change “Whilst” (British, archaic) to “While”.
The comment was noted and addressed.
22. Line 62: ….by a cis-acting element.
The comment was noted and addressed.
23. Line 63: …known as the displacement loop (D-Loop) region which also acts as the….
The comment was noted and the hyphen was removed.
24. Line 69/70: [?] Interestingly, no reports on mutations of mitochondrial transcription gene were observed in mitochondrial disease. [I do not understand what is meant by this sentence. What is a “mitochondrial transcription gene”? Do the authors mean a mtRNA Polymerase gene? (Which is not located on mtDNA.) Do they mean tRNA and rRNA genes? (Which would make the sentence false.)]
The concern of the reviewer was noted and the sentence was removed from the text.
25. Line 70/71: Most of the proteins present in mitochondria are [not is] encoded…..
The comment was noted and addressed.
26. Line 73: Dysfunction in mitochondrial replication results in…. [remove the word “process”].
The comment was noted and addressed.
27. Line 76/77: … during the neonatal phase or adolescence or, in some cases, during adult onset.
The comment was noted and addressed.
28. Line 77: … 1 in 4300 [or 4,300].
The comment was noted and the comma was added.
29. Line 78: “mitochondrial diseases” was used as a collective term for a heterogeneous group of genetic disorders characterized by…..
The comment was noted and addressed.
30. Line 80: ….and of nuclear-derived proteins.
The comment was noted and addressed.
31. Line 81: …in the lipid milieu, in mitochondrial translation, and in mitochondrial fission and fusion.
The comment was noted and addressed.
….. defects in the lipid milieu, in mitochondrial translation, and in mitochondrial fission and fusion
32. Line 85: ….features that can be categorized into……
The comment was noted and addressed.
33. Line 92: … or the risk of developing disease…..
The comment was noted and addressed.
34. Line 94: ….population as 11.82 per 100,000.
The comment was noted and addressed.
35. Line 95: …. population as 2.6 per 100,000.
The comment was noted and addressed.
36. Line 96 … on the prevalence of MELAS as 0.2 per 100,000.
The comment was noted and addressed.
37. Line 101: …. Various organs….
The comment was noted and addressed.
38. Line 102: …. It usually affects individuals [not It is usually affects…]
The comment was noted and the word ‘is’ is removed.
39. Line 106: Add space before [32].
The comment was noted and addressed.
40. Line 113: can manifest in both children and adults. [not manifests].
The comment was noted and addressed.
41. Line 113: Symptoms… include [not includes].
The comment was noted and addressed.
42. Line 122: a mutation of the mtDNA, a single amino acid substitution in one of the ….. [remove “which causing”, “substitution” is better than “exchange”]
The comment was noted and addressed.
43. Line 123: The majority of….
The comment was noted and addressed.
44. Line 128: pediatric [not paediatric (British)]
The comment was noted and addressed.
45. Line 137: …. which affects mitochondrial function, result in classic [not “results” (mutations result, not mutations results)]
The comment was noted and addressed.
46. Line 140: Mutations…. do not differ….. [ not mutations does not differ).
The comment was noted and addressed.
47. Line 144: NGS utilizes single stranded DNA [“utilizes” not “sequences”].
The comment was noted and addressed.
48. Line 146: DS allows >10,000-fold greater accuracy than conventional NGS [not 10,000-fold more accurate than conventional NGS].
The comment was noted and addressed.
49. Line 149/150: awkward sentence. I am not sure of the intent. Original sentence: “The first report on DS was a study to compare the mitochondrial genome mutation indentified in breast cancer epithelial cells with stem cell features and without.” Do they mean “The first report on DS was a study to compare the mitochondrial genome mutation indentified in breast cancer epithelial cells with and without stem cell features.”?
The comment was noted and the sentence was rephrased.
The first report on DS was a study investigating the mutational variations of the whole mitochondrial genome between non-stem cells and stem cells in human breast tissues.
50. Line 151/152: …. and that the majority of mutational variation occurred randomly.
The comment was noted and addressed.
51. Line 157/158: This method was concluded to be a robust, accurate and sensitive analysis, suitable for large samples.
The comment was noted and addressed.
52. Line 166: The transfer of exogenous [rather than “artificial”] mitochondria. Space before [53].
The comment was noted and addressed.
53. Line 167: mitochondrial network can lead to an alteration in the function…
The comment was noted and addressed.
54. Line 170: space before [54].
The comment was noted and addressed.
55. Line 172: into empty mitochondria. [remove “shells”].
The comment was noted and addressed.
56. Line 174: …only for a short duration of time.
The comment was noted and addressed.
57. Line 177: … permeability which increases the number of mitochondria penetrating….
The comment was noted and addressed.
58. Line 180: Various types…. Which include …..
The comment was noted and addressed.
59. Line 183/184: an increase in mitochondrial DNA concentration, OXPHOS activity, and ATP production….
The comment was noted and addressed.
60. Line 195: Biolistic transformation utilizes a helium shockwave to deliver DNA on microscopic metal particles….
The comment was noted and addressed
61. Line 198/199: …where biolistic transformation has been used to deliver DNA into organelles.
The comment was noted and addressed
62. Line 202: … and to repopulate with wild-type DNA.
The comment was noted and addressed
63. Line 212: …the goal of eliminating mitochondria …
The comment was noted and addressed
64. Line 224: … fibers that are associated with….
The comment was noted and addressed
65. Line 253: …. Are among the most common …. [Amongst (archaic)].
The comment was noted and addressed
66. Line 256: additional complications due to
The comment was noted and addressed
67. Line 257: homoplasmy is misspelled.
The comment was noted and spelling was corrected.
68. Line 264: suspected mitochondrial diseases. [delete “cases”].
The comment was noted and addressed

Round 2
Reviewer 1 Report
The revised manuscript addresses most of my comments and concerns with the incorporation of some text as well as the addition of two figures to illustrate the text.
Author Response
Comments
There are still some problems with the language which need to be improved, e.g. in the abstract:Pathogenic mitochondria'l' DNA mutations result in mitochondrial disorders...
The comment was noted and addresses
Please write always mitochondrial DNA disorders (or diseases), since the term mitochondrial disorders (diseases) is related to also to diseases due to nuclear gene mutations affecting mitochondrial proteins, which the review is not discussing. That should be also clearly mentioned under point 3. (Line 99 ff).
The terms ‘mitochondrial disease’ and ‘mitochondrial disorders’ are changed to ‘mitochondrial DNA disease’ and ‘mitochondrial DNA disorders’
I would recommend to avoid such unprecise statements (line 95): The basic mitochondrial translational machinery is encoded by mtDNA. Instead: Mitochondrial DNA encodes the full set of tRNAs and the two ribosomal RNAs required for mitochondrial translation.
The comment was noted and the sentence was replaced as per suggested.
Please correct Figure 2.
We acknowledge there is an error in Figure 2. However, we believe that this error maybe due to some technical error when the figure file and manuscript were merged. We seek the help of the Editorial team to resolve this issue.
